# Monoterpenes as Sirtuin-1 Activators: Therapeutic Potential in Aging and Related Diseases

**DOI:** 10.3390/biom12070921

**Published:** 2022-06-30

**Authors:** Cátia Sousa, Alexandrina Ferreira Mendes

**Affiliations:** 1Centre for Neuroscience and Cell Biology, University of Coimbra, 3004-504 Coimbra, Portugal; 2Faculty of Pharmacy, University of Coimbra, 3004-548 Coimbra, Portugal; 3Centre for Innovative Biomedicine and Biotechnology, University of Coimbra, 3004-504 Coimbra, Portugal

**Keywords:** aging, age-related diseases, monoterpenes, monoterpenoids, iridoids, secoiridoids, Sirtuin-1

## Abstract

Sirtuin 1 (SIRT) is a class III, NAD^+^-dependent histone deacetylase that also modulates the activity of numerous non-histone proteins through deacylation. SIRT1 plays critical roles in regulating and integrating cellular energy metabolism, response to stress, and circadian rhythm by modulating epigenetic and transcriptional regulation, mitochondrial homeostasis, proteostasis, telomere maintenance, inflammation, and the response to hypoxia. SIRT1 expression and activity decrease with aging, and enhancing its activity extends life span in various organisms, including mammals, and improves many age-related diseases, including cancer, metabolic, cardiovascular, neurodegenerative, respiratory, musculoskeletal, and renal diseases, but the opposite, that is, aggravation of various diseases, such as some cancers and neurodegenerative diseases, has also been reported. Accordingly, many natural and synthetic SIRT1 activators and inhibitors have been developed. Known SIRT1 activators of natural origin are mainly polyphenols. Nonetheless, various classes of non-polyphenolic monoterpenoids have been identified as inducers of SIRT1 expression and/or activity. This narrative review discusses current information on the evidence that supports the role of those compounds as SIRT1 activators and their potential both as tools for research and as pharmaceuticals for therapeutic application in age-related diseases.

## 1. Introduction

The 20th century brought about an unprecedented improvement in living conditions for a large fraction of the world population, which, along with the development of many new drugs, namely anti-microbial drugs, including vaccines, resulted in a significant reduction of childhood mortality and extension of longevity. Even though such improvements are highly unevenly distributed throughout the world, the outcomes are still impressive worldwide. Many diseases, however, are still incurable, and the increased lifespan is not yet accompanied by a parallel increase in health span. Indeed, as aging is the major risk factor for many diseases, increased longevity is associated with a higher incidence and thus prevalence of most chronic diseases from cardiovascular, neurodegenerative, metabolic, musculoskeletal, kidney, and respiratory diseases to cancer. These diseases pose a huge burden on individuals as well as on societies with increasing, soon-to-be unbearable costs [1,2].

Regardless of controversies concerning the definition and understanding of aging and its relation with disease, a consensus exists as to the need of urgent measures to promote healthier aging [2]. Among potential measures, pharmacological prevention of age-related disease development and progression is promising, as several drugs have been shown to slow down aging and extend lifespan in model organisms and/or to have beneficial effects in cell and animal models of age-related diseases. Several signaling pathways and molecular components have been identified as being targeted by those drugs that hence show promise for further development as treatments for age-related diseases and eventually as anti-aging drugs [3,4,5,6].

Among those, sirtuins (SIRTs), a family of NAD^+^-dependent deacylating enzymes, are of special interest. SIRTs, which in humans comprise seven members (SIRT1-7), are class III histone deacetylases (HDAC) that differ from other classes of the same family in that the deacetylation reaction occurs in two steps that consume nicotinamide adenine dinucleotide (NAD^+^), resulting in the formation of the deacetylated protein, nicotinamide, and 2′-O-acetyl-ADP-ribose [7]. Notwithstanding, different SIRTs can also remove other acyl groups, namely succinyl, malonyl, glutaryl, long-chain fatty acyl, and ADP-ribosyl groups, from many different non-histone proteins, including transcription factors and other enzymes [7]. Furthermore, SIRTs also differ as to the subcellular location, which determines their specific roles. SIRT1, 6, and 7 are nuclear enzymes although SIRT1 can also be found in the cytoplasm, whereas SIRT2 is mainly cytosolic, and SIRT3, 4, and 5 are mitochondrial enzymes [8,9].

SIRTs act as nutrient and metabolic sensors that, by responding to changes in the NAD^+^/NADH ratio, increase the ability of cells to cope with stress conditions. Therefore, SIRTs play fundamental roles in many cellular processes that link energy metabolism and stress responses, being involved in epigenetic and transcriptional regulation, mitochondrial homeostasis, proteostasis, telomere maintenance, inflammation, and the response to hypoxia. These processes regulate crucial cell functions from DNA repair to proliferation, survival, differentiation, and circadian rhythms, all of which are critically dysregulated in aging and age-related diseases [8,10,11,12]. Accordingly, SIRTs, in particular SIRT1 and 6, have been shown to increase life span in model organisms, including rodents [9,10,13,14].

The discovery of small molecule activators and inhibitors of SIRTs, particularly of SIRT1, further incremented research on the pathophysiologic roles of these enzymes and their potential therapeutic applications [15,16,17] even though controversies exist as to their protective or detrimental roles in various diseases [18]. The molecules first identified as activators of SIRT1 and other SIRTs are natural compounds belonging to the polyphenol family and include resveratrol, the most potent; curcumin; quercetin; and fisetin [14]. Starting from these, many derivatives have been designed attempting to overcome the low potency, bioavailability, and specificity of the parent compounds, while many fully synthetic compounds of various chemical classes have been developed with similar goals [15,16,17]. Other classes of natural compounds have been less studied regarding the ability to modulate SIRT1 and/or other SIRTs [19,20,21,22,23] even though many show pharmacological effects that coincide with those reported for polyphenols, which have been shown to be mediated, at least in part, by modulation of SIRT1 [24,25,26]. Therefore, non-polyphenolic natural compounds can represent an as.yet unappreciated source of chemicals for development of SIRT modulators useful to counteract aging and related diseases.

Here, we review and discuss current information on the evidence that supports the role of natural, non-polyphenolic, monoterpene compounds as SIRT1 activators and their potential both as tools for research and as pharmaceuticals for therapeutic applications in aging and related diseases. We will also highlight some controversies and hurdles that still persist about SIRT1 and its potential to slow down aging and age-related diseases.

## 2. Sirtuin-1: A Target to Offset Aging and Age-Related Diseases

Among the mammalian SIRTs, SIRT1 is the most studied, as it was the first to be identified as the mammalian ortholog of Sir2, a protein encoded by the *Silent Information Regulator 2* (*Sir2*) gene and the first NAD^+^-dependent deacetylase [27,28] found to extend lifespan in the yeast, *Saccaharomyces cerevisiae*, in response to stress conditions, namely to calorie restriction [29]. Shortly after, Sir2 homologs with similar functions were identified in other model organisms, including mammals, and shown to also extend life span and to mediate the positive effects of calorie restriction [30,31,32].

Besides longevity extension, SIRT1 has been attributed many beneficial effects in a variety of age-related diseases ranging from metabolic disturbances, such as obesity and diabetes mellitus, especially type 2, as well as cardiovascular, neurodegenerative, pulmonary, renal, musculoskeletal, autoimmune, and skin diseases to cancer [11,33,34,35,36,37,38,39,40,41]. The beneficial effects on life- and health-span mediated by increased SIRT1 expression and/or activity result from multiple mechanisms that involve the deacetylation of key lysine (K) residues in histones, including H3K9, H4K16, and H1K26, and numerous non-histone protein targets that include p53, forkhead box protein O1/3 (FOXO1/3), peroxisome proliferator-activated receptor gamma coactivator 1α (PGC-1α), 5′ AMP-activated protein kinase (AMPK), nuclear factor kappa-light-chain-enhancer of activated B cells (NF-κB), Period (Per) 1 and 2, and cryptochrome (Cry). By targeting these proteins, SIRT1 is able to regulate numerous vital signaling pathways and cellular processes, including DNA stability and repair, cell death, proliferation and differentiation, mitochondrial biogenesis and function, autophagy, and circadian rhythms. These processes and signaling pathways in turn regulate not only specific tissue and organ functions but also homeostatic mechanisms critical for whole body homeostasis, namely glucose and insulin homeostasis, hormone secretion, inflammation, cellular senescence, and central and peripheral circadian clocks (Figure 1).

Dysregulation of these mechanisms is associated with aging and related diseases, and increased SIRT1 expression and/or activity has been shown to improve life- and health-span, at least in part, by modulating those mechanisms [42,43,44]. In particular, SIRT1 has been shown to play a critical role in regulating and integrating cell metabolism; adaptation to stress, including inflammation and cell senescence; and circadian rhythms. Such integration involves the interplay between SIRT1, components of the clock system, and NF-κB, the master regulator of inflammation.

The interaction between SIRT1 and the clock system is complex. On one hand, SIRT1 and Per2 are involved in an antagonistic crosstalk in which SIRT1 deacetylates histone (H) 4 K16 in the Per2 promoter, decreasing its expression, while Per2 binds to the Sirt1 promoter, preventing BMAL1:CLOCK-induced transcription [45,46]. On the other hand, SIRT1 and CLOCK act coordinately but in opposite directions to regulate the clock system. CLOCK was shown to induce the rhythmic acetylation of BMAL1 and H3 K9/K14 at circadian promoters, which correlated with circadian oscillations of SIRT1 activity. Furthermore, SIRT1 association with CLOCK and recruitment to the BMAL1:CLOCK-chromatin complex at circadian promoters was shown to be indispensable for circadian control [47]. Another relevant mechanism of integration of metabolism and the clock system involves the circadian regulation of NAD^+^ levels, which in turn determine SIRT1 activity. NAD^+^ production depends on the expression and activity of the enzyme nicotinamide phosphoribosyltransferase (NAMPT or visfatin), which is rhythmically induced by the core complex BMAL1:CLOCK. Furthermore, SIRT1 associates with BMAL1:CLOCK at the NAMPT promoter, contributing to the rhythmic circadian expression of this enzyme and, therefore, to circadian oscillations of NAD^+^ levels and its own activity [48]. The relevance of these interactions has been demonstrated in a large number of studies that showed altered expression of clock genes and impaired circadian rhythms correlated with decreased SIRT1 levels in in vitro and in vivo models of aging and various diseases. SIRT1-deficient mice exhibit disturbed circadian rhythms and premature aging, whereas genetic or pharmacological interventions that increase SIRT1 levels and/or activity restore the circadian rhythm and counteract aging and disease manifestations [37,45,49,50,51].

SIRT1 and the clock system are also important regulators of the cellular response to stress by reciprocally modulating NF-κB. This transcription factor is the central regulator of the cellular response to stress, inflammation, and immune responses and plays a significant role in cell senescence, aging, and age-related diseases [52,53,54]. SIRT1 inhibits the transcriptional activity of NF-κB directly by deacetylating its RelA/p65 subunit at K310, a residue critical for transcriptional activity [55]. SIRT1 can also modulate NF-κB activity by indirect mechanisms that involve activation of negative regulators of this transcription factor, namely AMPK, PGC-1α, and PPAR-α [56]. NF-κB, in turn, can negatively modulate SIRT1 expression and activity, namely by inducing the expression of the microRNA, miR-34a, and increasing oxidative stress, which by multiple mechanisms can decrease SIRT1 expression and reduce its activity directly and indirectly by decreasing NAD^+^ levels [56,57]. Moreover, NF-κB-induced cytokines, such as TNF-α, have been shown to downregulate SIRT1 activity by inducing its proteolytic cleavage [58].

On the other hand, inflammatory signaling pathways and the circadian rhythm system are also intertwined, regulating each other in a complex manner. Inflammatory cytokines whose expression is induced by NF-κB, namely Interleukin (IL)-1β, have been shown to disrupt the rhythmic expression of clock genes and clock-controlled catabolic pathways relevant in age-related degenerative diseases. In response to acute and chronic inflammation, NF-κB was shown to directly interact with BMAL1:CLOCK, driving the complex away from the promoter regions of its primary target genes, namely Cry, Per, and Rev-erb genes, thus repressing the negative feedback loop of the circadian clock [59,60,61]. Taken together, these findings highlight the role of NF-κB on inflammation-induced circadian rhythm disruption, but in basal conditions, without inflammation, NF-κB was also shown to play a significant role in normal circadian oscillations of clock genes, being required for the normal circadian rhythmicity of mouse behavior [61].

Conversely, clock proteins were also found to regulate NF-κB activation and transcriptional activity. On one hand, CLOCK was shown to directly associate with NF-κB at the promoters of its target genes, increasing their transcription, probably by promoting RelA/p65 acetylation directly or through recruitment of other histone acetyltransferases. BMAL1 instead downregulates the CLOCK-dependent increase in NF-κB activity, likely by associating with CLOCK and driving it away from its interaction with NF-κB [62]. Thus, the availability of CLOCK and BMAL1 can be an important mechanism to finely tune NF-κB activity and daily variations in the intensity of the inflammatory response to a variety of stimuli. In contrast, RORα, a component of the negative feedback loop of the circadian clock, was found to downregulate NF-κB activity by two distinct mechanisms: (i) direct binding and transactivation of a ROR response element in the promoter of IκB-α, the major NF-κB inhibitor, increasing its expression [63], and (ii) binding to NF-κB target gene promoters and recruitment of HDAC3 that decreases RelA/p65 acetylation and transcriptional activity [64].

The interdependence of SIRT1, circadian clock components and NF-κB, highlights the physiologic relevance of their coordinated interaction so that metabolism, circadian rhythms, and the ability to cope with stress can be adequately integrated to maintain homeostasis. In turn, disruption of this balance is associated with aging and related diseases. Conversely, restoring SIRT1 expression and activity. which are decreased in aging and disease, is an attractive strategy to increase life- and health-span [12,16,17].

## 3. Monoterpenes and SIRTs

Terpenoids, also known as isoprenoids, are a family of natural products, particularly secondary metabolites, comprising a diversity of structures [65]. This family is classified based on the organization and number of isoprene units, which is considered the building block of terpenoids and corresponds to the molecular formula C_5_H_8_ [66]. The terms terpenoid and terpene are frequently used interchangeably. However, terpene refers to hydrocarbons with olefinic bonds, while terpenoids are terpenes with different functional groups, such as hydroxyl or ketone groups [67]. According to the number of isoprene units, terpenoids are divided into hemiterpenoids (C_5_H_8_), monoterpenoids (C_10_H_16_), sesquiterpenoids (C_15_H_24_), diterpenoids (C_20_H_32_), sesterpenoids (C_25_H_40_), triterpenoids (C_30_H_48_), tetraterpenoids or carotenoids (C_40_H_64_), and polyterpenoids ((C_5_H_8_)_n_) [65]. Monoterpenoids or monoterpenes consist of a 10-carbon backbone corresponding to two isoprene units and can be divided in acyclic, monocyclic, bicyclic, and irregular compounds. Moreover, iridoids are also considered monoterpene derivatives.

### 3.1. Monoterpenes

Monoterpenes are the main components of essential oils and have been ascribed a diverse range of pharmacological activities [68]. Nevertheless, whether these activities are a consequence of the effects of monoterpenes on SIRTs is not fully elucidated. There are only a few studies reporting the effects of these compounds on SIRTs, mainly on SIRT1.

#### 3.1.1. Bakuchiol

Bakuchiol (Table 1) is a monocyclic monoterpene phenol that has been reported to have antioxidant [69], anti-aging [70], and anti-inflammatory [71] properties. This compound is an analogue of resveratrol, having a resveratrol-like structure but differing from resveratrol in that it has only one hydroxyl group on the phenyl ring [72]. Since resveratrol inhibits ROS production in cardiomyocytes via SIRT1 activation, Feng and colleagues (2016) hypothesized that bakuchiol might protect tissues from myocardial ischemia-reperfusion injury (IRI) by attenuating oxidative stress via the SIRT1/PGC-1α pathway. *Ex vivo* studies using normal rat hearts treated with bakuchiol increased SIRT1 protein levels and, consequently, PGC-1α protein levels. Moreover, pre-treatment with bakuchiol counteracted the decrease of SIRT1 and PGC-1α protein levels induced by IR in rat hearts. As downstream effects, bakuchiol increased the activity of succinate dehydrogenase (complex II), cytochrome c oxidase (complex IV), and superoxide dismutase (SOD) as well as mitochondrial redox potential, whereas it decreased H_2_O_2_ formation and malondialdehyde content. Moreover, bakuchiol increased the levels of the anti-apoptotic factor, Bcl-2, and decreased the levels of the pro-apoptotic factors, Bax-2, and cleaved Caspase 3, thereby exhibiting cardioprotective effects against myocardial IRI. These results were confirmed using a known SIRT1 inhibitor, sirtinol, in IR-injured hearts and a SIRT1 small interfering RNA (siRNA) in IR-injured cardiomyocytes, respectively. These studies showed that decreasing SIRT1 activity and expression were both efficient in abolishing the protective effects of bakuchiol, suggesting that this terpenoid attenuates IRI by activating the SIRT1/PGC-1α pathway [73]. Corroborating these findings, Ma and co-workers (2020) reported that bakuchiol ameliorates hyperglycemia-induced diabetic cardiomyopathy by activating the SIRT1/Nrf2 signaling pathway. Using a streptozotocin-induced diabetes mouse model, the authors found that treatment with bakuchiol increases SIRT1 levels and its deacetylase activity as well as Nrf2 nuclear accumulation in the diabetic myocardium. Consequently, an improvement of cardiac function, mitigation of pathologic cardiac hypertrophy, prevention of myocardial fibrosis, and suppression of oxidative stress were observed. Accordingly, the administration of EX-527, a selective SIRT1 inhibitor, abolished all the beneficial effects of bakuchiol on diabetic hearts. Concordant with the in vivo results, bakuchiol restored SIRT1 protein levels, its deacetylation activity, and Nrf2 nuclear accumulation and prevented cell death and oxidative stress in a cardiomyocyte cell line subjected to high glucose (HG)-induced damage. The effects of bakuchiol were abolished by SIRT1 and Nrf2 siRNA treatment, but Nrf2 siRNA had no significant impact on SIRT1 levels or activity, implying that Nrf2 acts downstream to SIRT1 [74]. Overall, these studies point out bakuchiol as an inducer of SIRT1 expression [73,74].

#### 3.1.2. (S)-(+)-Carvone

(S)-(+)-carvone (Table 1) is a monocyclic ketone monoterpene with reported anti-inflammatory properties in various cell and animal models [75] that we recently found to directly increase SIRT1 activity without affecting its protein levels, suggesting that this compound is an allosteric activator of SIRT1 [21]. Accordingly, we observed that (S)-(+)-carvone decreased lipopolysaccharides (LPS)-induced acetylation of the NF-κB subunit, p65, at K310, which resulted in reduced expression of inflammatory mediators, such as IL-1β and inducible nitric oxide synthase, in murine macrophages and human chondrocytes [76].

#### 3.1.3. Hinokitiol

Hinokitiol (Table 1), also known as β-thujaplicin, is a monocyclic tropolone monoterpene that has shown various pharmacological activities, such as anti-microbial, anti-cancer, and anti-inflammatory [77]. Regarding the anti-inflammatory properties, Lee and colleagues (2017) demonstrated the ability of hinokitiol to decrease several inflammation markers, including TNF-α, IL-6, prostaglandin E2, and matrix metalloproteinase (MMP) 9 secretion and cell migration while inhibiting NF-κB nuclear accumulation in LPS-treated normal human keratinocytes (NHEKs). Moreover, hinokitiol increased SIRT1 protein levels and deacetylase activity, while it decreased p53 acetylation. The role of SIRT1 in mediating those anti-inflammatory effects was confirmed with a SIRT1 inhibitor, sirtinol, and a SIRT1, siRNA, both of which abolished all the effects mediated by hinokitiol. Conversely, overexpression of SIRT1 mimicked the effects observed using hinokitiol [78]. Since hinokitiol increased SIRT1 protein levels, and its siRNA abolished its anti-inflammatory effects, it is likely that this compound acts as an inducer of SIRT1 expression. Nonetheless, since it was not evaluated, a direct increase of enzyme activity cannot be excluded as an underlying mechanism contributing to the anti-inflammatory effects observed.

#### 3.1.4. Paeoniflorin

Paeoniflorin (Table 1) is a pinane glycoside monoterpene [79] with reported beneficial effects in models of cardiovascular, neurological, and renal diseases [80]. Moreover, paeniflorin was reported to attenuate pain caused by chronic constriction injury (CCI) of the sciatic nerve in rats in association with increased protein levels of brain-derived neurotrophic factor (BDNF) [81], a growth factor that plays a crucial role as a neuromodulator of pain transmission both in the peripheral and central nervous systems that also has anti-inflammatory properties [82]. In the study by Tao et al. (2017), the anti-nociceptive effect of paeoniflorin was associated with up-regulated mRNA and protein levels of SIRT1 and CREB, which were downregulated in the spinal dorsal horn and dorsal root ganglion by CCI, suggesting that its positive effects were possibly due to the induction of SIRT1 expression [81]. Another study reported that by enhancing autophagy, paeoniflorin inhibited apoptosis and the expression of adhesion molecules, namely E-selectin, VCAM, and ICAM, induced by oxidized low-density lipoprotein (ox-LDL) in human umbilical vein endothelial cells (HUVECs). Moreover, paeoniflorin induced SIRT1 protein levels in the absence and presence of ox-LDL in HUVECs, and EX-527 reversed its effects in autophagy, apoptosis, and adhesion molecule expression. Thus, this study suggests that paeoniflorin promotes its beneficial effects via SIRT1 up-regulation [83].

Collectively, these studies point out paeoniflorin as an inducer of SIRT1 expression.

**Table 1 biomolecules-12-00921-t001:** Monoterpenes and SIRT1.

Compound	Structure	Model and (Dose or Concentration of Test Compound)	SIRT1		Outcome	Ref.
			Activity	Expression		
Bakuchiol	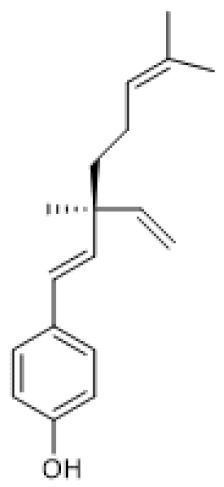	Normal Rat Heart (0.25–1 µM) IR-Injured Rat Heart (0.25–1 µM)	NA	↑	Attenuation of myocardial IR injury	[73]
		STZ-induced mouse (60 mg/Kg/d) HG-treated H9c2 cells (10 µM)	NA	↑	Alleviation of hyperglycemia-induced cardiomyopathy	[74]
(S)-(+)-carvone	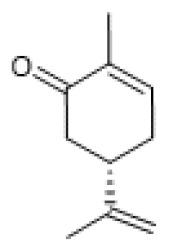	LPS-stimulated Raw 264.7 cells	↑	=	Reduction of LPS-induced pro-inflammatory mediators	[21]
Hinokitiol	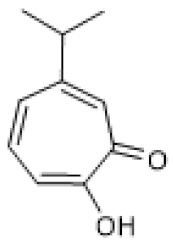	LPS-stimulated NHEKs cells (2.5–20 µmol/L)	NA	↑	Attenuation of LPS-induced inflammation	[78]
Paeoniflorin	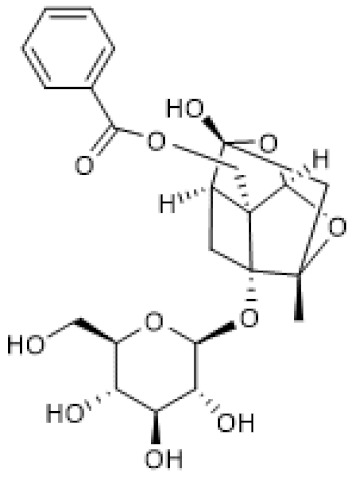	chronic constriction injury of rat sciatic nerve (50–100 mg/kg)	NA	↑	Attenuation of neuropathic pain	[81]
		Ox-LDL-treated HUVEC cells (100 µg/mL)	NA	↑	Attenuation of Ox-LDL-induced apoptosis and adhesion molecule expression	[83]

HG, high glucose; HUVEC, human umbilical vein endothelial cells; IR, ischemia reperfusion; LPS, lipopolysaccharides; NA, not applicable; NHEK, normal human epidermal keratinocyte; ox-LDL, oxidized low-density lipoprotein; STZ, streptozotocin; =, no alteration.

### 3.2. Iridoids

Belonging to the monoterpenoid class, iridoids are also characterized by the 10-carbon backbone in which a six-membered ring with an oxygen atom is fused to a cyclopentane ring, the iridane skeleton. These compounds are frequently combined with sugar moieties, forming glycosides. They are divided in four groups: non-glycoside or simple iridoids, iridoid glycosides, secoiridoids that result from cleavage of a bond in the cyclopentane ring, and bisiridoids, which result from the dimerization of iridoids and secoiridoids [65,84].

#### 3.2.1. Non-Glycoside Iridoids

Genipin (Table 2) is an aglycone derived from geniposide, an iridoid glycoside. Diverse therapeutic effects have been attributed to genipin, namely anti-tumor and hepatoprotective properties [85]. Shin and co-workers (2017) investigated the mechanisms underlying the hepatoprotective effects of genipin, giving special attention to mitochondria. Using an in vivo model of hepatic ischemia reperfusion, the authors demonstrated that genipin attenuated the disruption of the mitochondrial structure and hepatocellular damage by reducing oxidative mitochondrial damage. Moreover, this compound counteracted the hepatic ischemia-reperfusion-induced decrease of mtDNA copy numbers, PGC-1α, Nuclear Respiratory Factor 1 (NRF1) and Transcription Factor A Mitochondrial (TFAM) protein levels, suggesting that genipin enhances mitochondrial biogenesis. Genipin also demonstrated the ability to regulate the imbalance in mitochondrial dynamics induced by ischemia-reperfusion by attenuating the alteration in the protein levels of dynamin-related protein (Drp1) and mitofusin 2 (Mfn2) induced by hepatic ischemia-reperfusion. Additionally, hepatic ischemia reperfusion induced the increase of mitochondrial PINK1, LC3-II, and p62 protein levels and the decrease of mitochondrial Parkin protein levels, suggesting mitochondrial damage and mitophagy impairment, respectively, that were restored by genipin. These positive effects of genipin on mitochondrial quality control mechanisms were related to its ability to attenuate the decrease in SIRT1 and phospho-AMPK levels induced by hepatic ischemia reperfusion. To confirm the involvement of AMPK/SIRT1 signaling, sirtinol was used and reversed the positive effects of genipin in hepatocellular and mitochondrial damage as well as in protein levels of mitochondrial Parkin and hepatic PGC-1α, suggesting that genipin promotes mitochondria quality control processes via the SIRT1 and AMPK pathway by increasing SIRT1 expression [86].

#### 3.2.2. Iridoid Glycosides

Catalpol (Table 2) has been extensively studied, and pharmacological effects, such as anti-inflammatory and analgesic, have been reported [87]. For instance, treatment with catalpol for 14 days ameliorated colitis symptoms, oxidative stress, endoplasmic reticulum (ER) stress, and apoptosis in a rat colitis model induced by intracolonic administration of 2,4,6-trinitrobenzene sulfonic acid (TNBS) [88]. Moreover, catalpol decreased the levels of miR132 and the acetylation levels of Heat Shock Factor protein 1 (HSF1), while it increased SIRT1 protein levels in the rat colitis model. Furthermore, catalpol inhibited endoplasmic reticulum (ER) stress induced by Brefeldin A in Caco-2 cells, and the treatment with SIRT1 siRNA abolished this effect. Moreover, catalpol was unable to increase SIRT1 protein levels in Brefeldin A-induced ER stress in Caco-2 cells in the presence of SIRT1 siRNA. Using miR132 mimics, the authors demonstrated that this microRNA targets SIRT1, downregulating its protein levels and, consequently, increasing the acetylation of HSF1, while catalpol reversed these effects. Thus, the results suggest that catalpol ameliorates colitis by modulating the miR132/SIRT1/HSF1 signaling pathway [88]. Corroborating these findings, Zhang and co-workers (2019) showed that catalpol exhibits protective effects against adriamycin-induced nephropathy by increasing SIRT1 mRNA and protein levels and, consequently, decreasing pro-inflammatory cytokines, namely IL-6 and TNF-α, cellular ROS accumulation, and apoptosis. Moreover, this compound down-regulated the levels of TRPC6 channels, which mediate the release of cytosolic compartmentalized calcium and up-regulated MRP2 expression to reduce adriamycin accumulation via SIRT1. These catalpol effects were abrogated by SIRT1 siRNA and EX-527. The effect of catalpol on SIRT1 was also corroborated by docking studies that demonstrated a strong affinity for SIRT1. This result suggests that catalpol can be a direct SIRT1 activator, but further studies are needed to verify this hypothesis [89]. Another study reported that catalpol ameliorates hyperglycemia, oxidative stress, and inflammation as well as associated pyroptosis in high-fat diet (HFD)/streptozotocin (STZ)-induced diabetic nephropathy in mice and high-glucose (HG)-induced podocyte injury model [90]. According to the authors, these effects were consequence of increased AMPK phosphorylation and SIRT1 protein levels and the decrease of NF-κB and inflammasome NLPR3 activation [90]. Using other disease models, Liu and colleagues (2021) demonstrated that catalpol increases SIRT1 protein levels in vitro and in vivo. Catalpol-induced SIRT1 up-regulation decreased oxidative stress and pro-inflammatory mediators by suppressing NF-κB and MAPKs signaling pathways in both imiquimod (IMQ)-induced psoriasis-like mice and TNF-α-stimulated human keratinocytes, HaCat, cells. Moreover, treatment with catalpol ameliorated IMQ-induced psoriasis-like phenotypic changes and symptoms in mice [91]. Overall, these studies suggest that the beneficial effects of catalpol are due to increased protein levels of SIRT1. Whether that increase occurs at the gene expression or other levels remains to be elucidated.

Geniposide (Table 2) is a well-known iridoid glycoside whose pharmacological effects have been widely described and include cardioprotective and anti-diabetic properties [92]. Treatment with geniposide was shown to increase SIRT1 mRNA and protein levels as well as its activity in mice challenged with isoproterenol (ISO), which causes cardiac fibrosis. Consequently, geniposide treatment significantly improved cardiac dysfunction, fibrosis, and hypertrophy. Moreover, using an in vitro model, namely TGF-β-treated cardiac fibroblasts, geniposide also increased SIRT1 mRNA and protein levels, which were decreased by TGF-β. To confirm these findings, EX-527 was used in both models and demonstrated that the beneficial effects of geniposide were partially mediated by SIRT1, being responsible for the inhibition of oxidative stress, ER stress, and Smad3 acetylation, while suppression of Smad3 phosphorylation by geniposide was independent of SIRT1 [93]. Still, regarding the cardioprotective effect of geniposide, Ma and colleagues (2018) also demonstrated that treatment with geniposide increased SIRT1 protein levels and, consequently, its activity in the heart, which were decreased in a model of obesity induced by feeding mice with a high-fat diet. The authors also showed that geniposide alone increased SIRT1 protein levels in cultured cardiomyocytes. Moreover, *Sirt1* knockdown abolished geniposide-mediated anti-inflammatory effects in palmitic acid-treated cardiomyocytes and hearts of obese mice. Interestingly, the authors found that geniposide did not increase SIRT1 protein levels in cardiomyocytes with inhibition of Glucagon-Like Peptide-1 Receptor (GLP-R1) or *Glp-1r* deficiency, suggesting that activation of SIRT1 by geniposide is mediated by the GLP-R1 [94] and thus that geniposide is a GLP-R1 agonist [95].

Another study showed that geniposide is able to reduce the development of STZ-induced diabetic nephropathy by inhibiting the NF-κB pathway [96]. This protective effect was later attributed to the ability of geniposide to block oxidative stress, inflammation, and associated pyroptosis by increasing total protein and phosphorylation levels of AMPK and SIRT1 protein levels and blocking NF-κB and NLRP3 inflammasome activation in both HFD/STZ-induced diabetic nephropathy in mice and HG-induced podocyte injury models [97]. Moreover, using an AMPK siRNA, SIRT1 protein levels were decreased in HG-treated podocytes, and a high concentration of geniposide counteracted this effect [97].

Another study showed that geniposide alleviates inflammatory responses in colitis via the AMPK/SIRT1 signaling pathway. Accordingly, geniposide increased the protein levels of SIRT1 in both in vivo and in vitro models, namely dextran sodium sulfate (DSS)-induced acute colitis in mice and LPS+ATP-treated murine bone-marrow-derived macrophages and the Raw 264.7 macrophage cell line. Interestingly, association of geniposide with EX-527 attenuated its effects, whereas association with an SIRT1 activator, SRT-1720, reinforced the beneficial effects described in the models used. Moreover, the same results were obtained when SIRT1 was knocked-down or overexpressed in Raw 264.7 cells [98]. Altogether, these studies demonstrated that the beneficial effects of geniposide are mediated by increasing the availability of SIRT1.

Genipin-1-β-D-gentiobioside (Table 2) has a chemical structure similar to that of geniposide except that it possesses one more glycosidic group. However, the pharmacological properties of genipin-1-β-D-gentiobioside have not been elucidated as extensively as those of geniposide. Genipin-1-β-D-gentiobioside efficiently suppressed inflammation and oxidative stress, thus improving renal function in a model of diabetic nephropathy, partly via AMPK/SIRT1 activation and NF-κB inhibition. As geniposide, genipin-1-β-D-gentiobioside increased AMPK protein and phosphorylation levels as well as SIRT1 protein levels, thus blocking NF-κB and NLRP3 inflammasome in both HFD/STZ-induced diabetic nephropathy in mice and HG-induced podocyte injury models [99], suggesting that the mechanism underlying these effects of genipin-1-β-D-gentiobioside is identical to that suggested for geniposide.

Loganin (Table 2) is another iridoid glycoside reported to have beneficial effects in a in vivo colitis model in relation with anti-inflammatory effects in LPS-treated human intestinal cell lines [100]. Moreover, loganin increased SIRT1 mRNA and protein levels in colon tissues of DSS-induced ulcerative colitis in mice, which were abolished by EX-527 [101]. Accordingly, loganin decreased RelA/p65 acetylation of lysine 310 and the expression of inflammatory mediators, namely IL-1β and IL-6, which were reversed by EX-527 treatment. Thus, the authors suggested that loganin attenuates DSS-induced colitis by modulating SIRT1/NF-κB pathway [101]. As with the other iridoid glycosides, loganin seems to exert its beneficial effects by inducing SIRT1 expression.

Monotropein (Table 2) was also reported as capable of increasing SIRT1 protein levels that were downregulated by H_2_O_2_ in primary osteoblasts. This effect was associated to decreased apoptosis and protein levels of pro-inflammatory mediators, namely cyclooxygenase 2, inducible nitric oxide synthase, IL-1β, IL-6, and TNF-α, in relation with decreased NF-κB nuclear accumulation [102].

Picroside II (Table 2) is another iridoid glycoside that was described as an inducer of SIRT1 expression [103]. Indeed, picroside-II increased SIRT1 levels in hyperhomocysteinaemic mice and homocysteine (Hcy)-treated HUVECs, and those effects were inhibited by treatment with EX-527 or SIRT1 siRNA, respectively. By increasing SIRT1 levels, picroside-II reduced lectin-like ox-LDL receptor-1 and endothelial damage [103].

Lastly, sweroside (Table 2) was described to increase SIRT1 protein levels that were downregulated by LPS in Raw 264.7 murine macrophages. Using nicotinamide as a SIRT1 inhibitor, the authors found that the effects of sweroside, including the increase in SIRT1 levels and decreased production of inflammatory mediators, were partially reversed [104].

**Table 2 biomolecules-12-00921-t002:** Non-glycoside and glycoside iridoids that increase SIRT1 expression.

Compound	Structure	Model and (Dose or Concentration of Test Compound)	SIRT1 Expression	Outcome	Ref.
Genipin	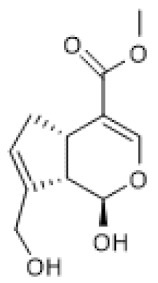	IR-induced hepatic mice injury (100 mg/Kg)	↑	Protection against IR-induced hepatic injury	[86]
Catalpol	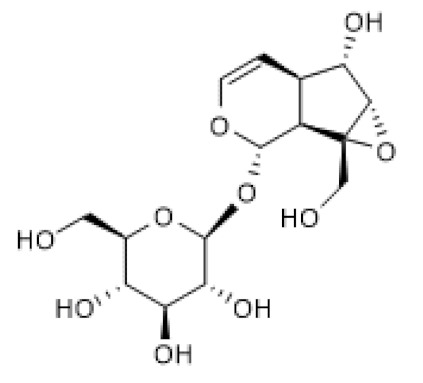	TNBS-induced rat colitis (20 mg/kg) Brefeldin A-treated Caco 2 cells (10–80 µM)	↑	Attenuation of ER stress in colitis	[88]
		Adriamycin-induced mice nephrotoxicity (40–120 mg/Kg)	↑	Alleviation of adriamycin-induced nephrotoxicity	[89]
		HFD/STZ-induced diabetic nephropathy mice (100–200 mg/Kg) HG-induced podocyte (10 µM)	↑	Inhibition of oxidative stress and inflammation	[90]
		IMQ-induced psoriasis-like lesions in mice (2.5–10 mg/Kg) TNF-α-stimulated HaCat cells (7.5–30 µM)	↑	Amelioration of psoriasis-like phenotypes	[91]
Geniposide	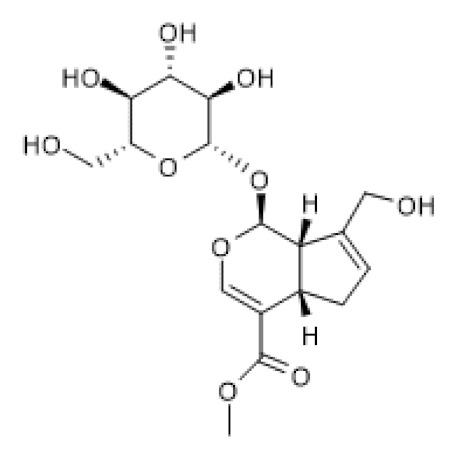	ISO-induced cardiac fibrosis in mice (50 mg/Kg/d) TGF-β-treated cardiac fibroblasts (100 µM)	↑	Alleviation of ISO-induced cardiac fibrosis	[93]
		HFD-fed mice (50 mg/Kg) Cardiomyocytes (50 µmol/L)	↑	Protection against obesity-related cardiac injury	[94]
		HFD/STZ-induced diabetic nephropathy in mice (25–50 mg/Kg) HG-induced podocyte (200 µg/mL)	↑	Alleviation of diabetic nephropathy	[97]
		DSS-induced acute colitis in mice (25–100 mg/Kg) LPS+ATP-stimulated Raw 264.7/BMDM cells (25–100 µM)	↑	Amelioration of inflammatory responses in colitis	[98]
Genipin- 1-β-D-gentiobioside	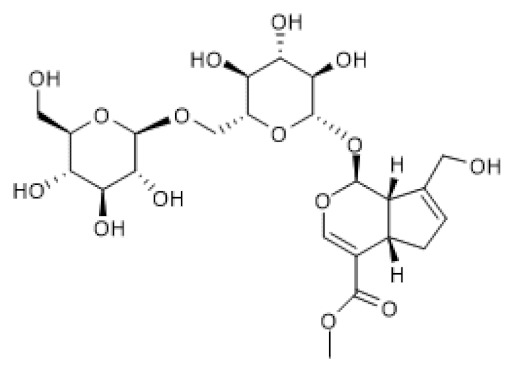	HFD/STZ-induced diabetic nephropathy in mice (25–50 mg/Kg) HG-induced podocyte (20 µM)	↑	Protection against diabetic nephropathy	[99]
Loganin	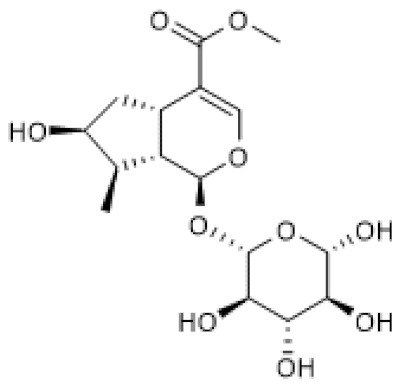	DSS-induced ulcerative colitis in mice (50–100 mg/Kg/day)	↑	Attenuation of DSS-induced ulcerative colitis	[101]
Monotropein	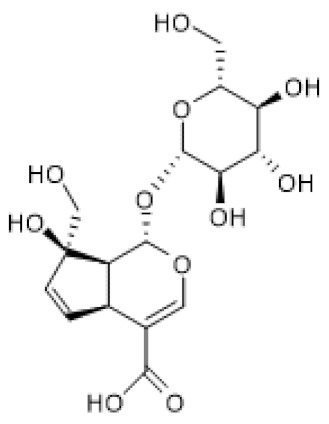	H_2_O_2_-treated primary osteoblasts (1.5–10 µg/mL)	↑	Suppression of apoptosis and inflammation in H_2_O_2_-treated primary osteoblasts	[102]
Picroside II	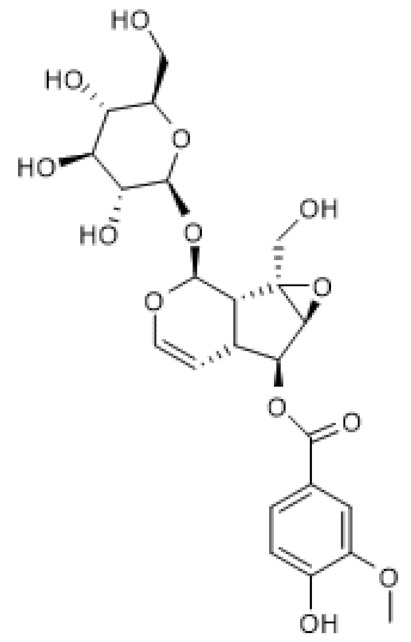	Hyperhomo-cysteinemic mice (60 mg/Kg) Hcy-treated HUVECs (100–200 µg/mL)	↑	Attenuation of hyperhomo-cysteinemia-induced endothelial injury	[103]
Sweroside	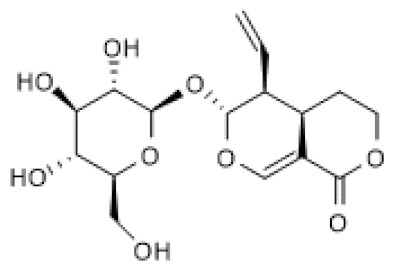	LPS-stimulated Raw 264.7 cells (40–80 µM)	↑	Attenuation of LPS-induced inflammation	[104]

DSS, dextran sulfate sodium; Hcy, homocysteine; HFD, high-fat diet; HG, high glucose; IMQ, imiquimod; ISO, isoproterenol IR, ischemia reperfusion; LPS, lipopolysaccharides; STZ, streptozotocin; TNBS, 2,4,6-trinitrobenzene sulfonic acid.

#### 3.2.3. Secoiridoids

Gentiopicroside (Table 3) is a secoiridoid glycoside for which several pharmacological activities have been described [105]. Zhou and colleagues (2021) reported the therapeutic effect of gentiopicroside on ovalbumin (OVA)-induced allergic asthma in mice by interfering with the NF-κB/SIRT1 signaling pathway. Accordingly, gentiopicroside increased SIRT1 and decreased acetylation of NF-κB/p65 protein levels in lung tissue of OVA-sensitized mice and decreased the levels of pro-inflammatory mediators in the bronchoalveolar lavage fluid, thus alleviating the pathological changes in lung tissue of OVA-sensitized mice. The effects induced by treatment with gentiopicroside were identical to those observed with resveratrol and partially reversed by treatment with EX-527 [106]. These results suggest that the anti-allergic effects of gentiopicroside are due in part to its ability to induce SIRT1 expression and thus inhibit NF-κB. Nonetheless, since these effects were only partially reversed by EX-527, it is probable that other mechanisms also contribute to the anti-inflammatory and anti-asthmatic activities of gentiopicroside.

Oleuropein (Table 3) is an ester of elenolic acid and hydroxytyrosol associated to a glycosidic residue, thus belonging to the class of secoiridoid glycosides. A variety of pharmacological activities have been attributed to oleuropein, including anti-inflammatory and autophagy inducing [107]. Oleuropein was described to increase SIRT1 mRNA levels and suggested to induce mitochondrial biogenesis via PGC-1α, whose mRNA levels were increased in cultured avian muscle cells treated with this compound [108]. Furthermore, oleuropein reduced mitochondrial superoxide anion generation, probably by increasing uncoupling protein (UCP) and manganese superoxide dismutase (MnSOD) mRNA levels [108]. These findings were confirmed under natural conditions using live animals, showing that oleuropein increases SIRT1 mRNA levels in the breast muscle of growing broiler chickens, leading to increased expression of antioxidant mitochondrial enzymes, transcription factors, and biogenesis-inducing factors, thus promoting mitochondrial function and reduced ROS production [109].

Similar to oleuropein, its aglycone also increased SIRT1 protein levels in 6-month-old TgCRND8 mice and in the neuronal cell line, N2a, treated with N-methyl-N′-nitro-N-nitrosoguanidine (MNNG), in in vivo and in vitro models of Alzheimer’s disease, respectively. Interestingly, oleuropein aglycone also decreased PolyADP-ribosylated proteins, which are the result of poly(ADP-ribose) polymerase 1 (PARP1) activation, in both models. Moreover, this compound also decreased PARP1 mRNA and protein levels in 6-month-old TgCRND8 mice, suggesting that oleuropein aglycone attenuated neuronal damage, at least in part, via modulation of the PARP1/SIRT1 signaling pathway [110]. PARP1-mediated cell death is a common feature of neurodegenerative diseases, including Alzheimer’s disease [111], and since both PARP1 and SIRT1 activity depend on the availability of NAD^+^ [112], the increase in SIRT1 levels induced by oleuropein aglycone likely contributes, at least in part, to counteract PARP1-induced cell death.
biomolecules-12-00921-t003_Table 3Table 3Secoiridoids that increase SIRT1 expression.CompoundStructureModelSIRT1 ExpressionOutcomeRef.Gentio-picroside
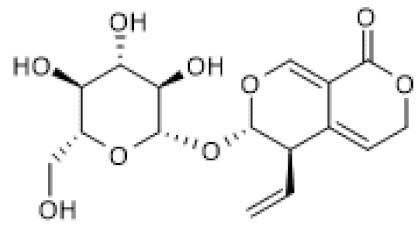
OVA-induced allergic mouse asthma (20–80 mg/Kg)↑Amelioration of OVA-induced inflammation[106]Oleuropein
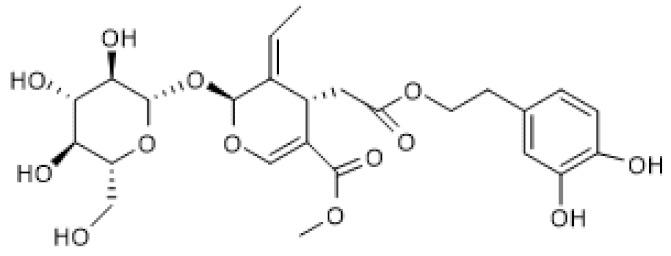
Avian muscle cells (50 µM)↑Induction of mitochondrial biogenesis and decrease of ROS[108]Growing broiler chicken (0.1–2.5 ppm)↑Reduction of muscle oxidative damage[109]Oleuropein aglycone
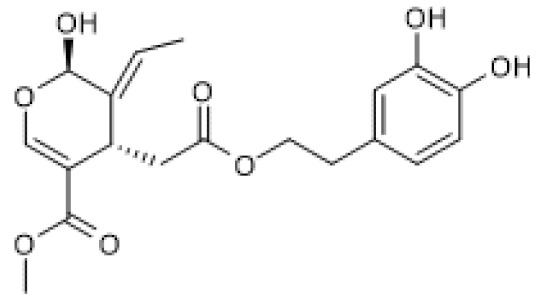
TgCRND8 mice(50 mg/Kg) MNNG-treated N2a cells (100 µM)↑Counteraction of neuronal damage[110]MNNG, N-methyl-N′-nitro-N-nitrosoguanidine; OVA, ovalbumin.

## 4. Conclusions

Multiple studies showed that increasing SIRT1 expression with the consequent increase in its protein levels and activity or enhancing its activity by allosteric interaction are efficient ways of increasing life- and health-span and to reverse the signs and symptoms of many age-related diseases [15,16,17,26]. Among many distinct natural and synthetic compounds that promote SIRT1 activity, monoterpenoids can also promote beneficial effects in a variety of pathological conditions by increasing SIRT1 activity. Nevertheless, unlike other classes of natural compounds known to act through SIRT1-mediated effects, the monoterpenoids seem to act mostly by inducing SIRT1 expression and, consequently, its activity, as indicated by the studies reviewed above. Thus, the molecular target of these molecules is not SIRT1 *per se* but other biomolecules that direct or indirectly regulate its expression. Among those, recent studies identified several non-coding RNAs, including microRNAs (miRNAs), long non-coding RNAs (lncRNAs), and circular RNAs (circ-RNAs), as modulators of SIRT1. Several micro RNAs (miRNAs) have been shown to decrease SIRT1 expression by targeting its mRNA both in disease models and in human diseased tissues compared to normal controls. miR-211 is one of those miRNAs recently shown to be downregulated in lesioned epidermis samples from vitiligo patients in relation with increased expression of a lncRNA, MALAT1, which sequesters miR-211, enhancing SIRT1 expression. In human keratinocytes, MALAT1-dependent miR-211 downregulation increased SIRT1 expression, which, in turn, promoted keratinocyte differentiation and protected the cells from UVB-induced DNA damage [113]. Another study showed that circ-SIRT1, a circ-RNA derived from the SIRT1 transcript, increases SIRT1 expression by acting as a cytoplasmic sponge for miR-132/211 while directly associating with NF-κB/p65 in the cytoplasm and preventing its nuclear translocation induced by TNF-α in vascular smooth muscle cells. Together, these mechanisms lead to increased SIRT1 expression and activity, resulting in decreased NF-κB/p65 acetylation and transcriptional activity, which decrease vascular inflammation [114]. These and other studies confirm the potential of targeting miRNAs to increase SIRT1 expression and activity, and recent studies also confirm that different natural compounds can target those miRNAs to increase SIRT1 expression. For instance, pterostilbene, a natural polyphenol, decreases liver fibrosis by increasing SIRT1 expression through downregulation of miR-34a [115]. Interestingly, catalpol, a monoterpene of the iridoid glycoside family described above, was shown to alleviate colitis in mice by inducing SIRT1 expression through inhibition of miR-132 [88], thus suggesting that non-polyphenolic monoterpenes can also modulate miRNAs, directly or indirectly, to increase SIRT1 expression. Thus, miRNAs and other non-coding RNAs are novel potential targets of non-polyphenolic monoterpenes that deserve to be investigated. Other biomolecules may also be involved in mediating the ability of monoterpenes to induce SIRT1 expression, and their identification is also essential not only to understand their mechanism of action and so design novel strategies to modulate aging and disease but also to predict potential adverse effects resulting from those molecular interactions.

On the other hand, most studies evaluated the effects of monoterpenes on SIRT1 at the mRNA and protein levels but did not evaluate their effects on the purified enzyme activity. Therefore, further studies are required to determine whether and which monoterpenes can directly increase SIRT1 activity by allosteric modulation in a mechanism similar to that involving SIRT1 activation by another class of natural compounds, the polyphenols, as well as by synthetic compounds [15,16,17].

In this respect, the observation that (S)-(+)-carvone directly enhances SIRT1 activity in chemico [21] demonstrates that non-polyphenolic monoterpenes also have the potential to allosterically interact with SIRT1 to increase its enzyme activity.

Despite the beneficial effects associated with SIRT1 activity, there are still controversies regarding its therapeutic potential. On one hand, there are conditions in which SIRT1 inhibition seems to be preferable, such as cancer and Huntington’s disease [16,116,117,118,119]. On the other hand, the beneficial effects of monoterpenes mediated by SIRT1 activation, namely increased cellular antioxidant capacity and improved mitochondrial biogenesis, may be hindered by the antioxidant properties that many of those compounds also present [120]. Indeed, it was shown that treatment with antioxidant vitamins abrogates or, at least, blunts the beneficial effects of calorie restriction and exercise training [121], two interventions well-known to increase SIRT1 and ameliorate aging and related diseases [122,123]. In agreement with the concept of mitohormesis, increasing evidence suggests that a small degree of oxidative stress may be required to potentiate the antioxidant response and improve bioenergetics so that the presence of an exogenous source of antioxidants eliminates the drive to activate the endogenous response to challenging conditions, such as exercise training and calorie restriction, which includes and is mediated, at least in part, by SIRT1. Whether monoterpenes can also compromise this hormetic response is unknown, but they may differ from other antioxidants, especially from free radical scavengers in that, besides some scavenging activity, they can also modulate the redox system by increasing the expression of antioxidant enzymes through activation of the transcription factor, nuclear factor E2-related factor 2 (Nrf2) [124,125].

From another point of view, potential deleterious consequences of increasing SIRT1 expression and activity also need to be considered. Upregulation of SIRT1 has been associated with various cancers [119], but the opposite, that is, a protective role of SIRT1 against some cancers, has also been reported [126]. For instance, as mentioned above, the axis MALAT1/miR-211/SIRT1 is increased in vitiligo keratinocytes, protecting these cells from UVB-induced DNA damage and promoting their differentiation which correlate with a lower incidence of melanomas and other cancers in vitiligo patients [113]. In the context of aging, upregulation of SIRT1 may also have detrimental effects. Increased SIRT1 serum levels have been associated with frailty and decreased physical function in older adults [127]. Although no direct cause-effect relationship was established, the authors of this study suggested that the low body mass index observed in frail individuals is paralleled by the effects of caloric restriction which decreases body weight while increasing SIRT1. Accordingly, low serum SIRT1 levels were found in less-frail elderly with higher total lean body mass [128], indicating that serum SIRT1 levels may reflect the nutritional status of an individual. Nonetheless, the source of serum SIRT1 has not been identified, and it is also conceivable that increased SIRT1 serum levels are due to release from damaged cells in different tissues affected by frailty. Among those, the skeletal muscle is especially relevant, as sarcopenia is a major feature of frailty. Interestingly, frailty and sarcopenia are associated with oxidative stress and chronic, low-grade inflammation [129,130], both of which are counteracted by SIRT1. Thus, increased SIRT1 may be the result of a failed attempt to overcome those processes in as much as a low level of oxidative stress promotes a hormetic response but becomes harmful when excessive. Clearly, much research is needed to fully elucidate the role of SIRT1 in health, aging, and disease. The availability of different classes of SIRT1 modulators, including monoterpenes, will help in dissecting the underlying mechanisms, which, in turn, will define their therapeutic utility. In summary, the identification of novel SIRT1 activators and inhibitors, such as (S)-(+)-carvone and eventually other non-polyphenolic monoterpenes, can contribute to solve the contradictions that still persist regarding the therapeutic potential of increasing or decreasing SIRT1 activity either directly or through modulation of its expression and can also provide new chemical scaffolds for the development of novel SIRT1 modulating compounds with improved specificity, potency, and pharmacokinetic properties.

## Figures and Tables

**Figure 1 biomolecules-12-00921-f001:**
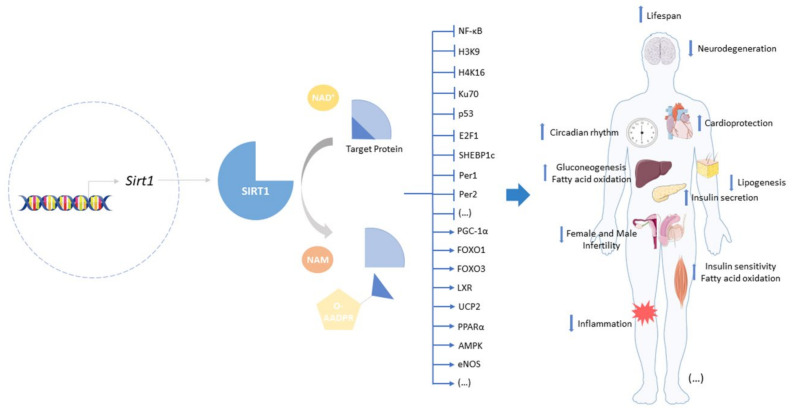
Deacetylation reaction, targets, and physiological consequences of SIRT1 activation. SIRT1 catalyzes the deacetylation of histone and non-histone target proteins by breaking the bonds between NAD^+^ and niacinamide, transferring the acetyl group from proteins to ADP-ribose, and releasing the deacetylated protein. Deacetylation can inhibit (ꓕ) or activate (↓) the target protein that will result in beneficial effects, such as longevity extension, neuroprotection, and cardioprotection.

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
