# Peer review of "Monoterpenes as Sirtuin-1 Activators: Therapeutic Potential in Aging and Related Diseases"

_biomolecules, 2022, doi:10.3390/biom12070921_

Round 1

Reviewer 1 Report

The review deals with monoterpenes as activators of SIRT1. The review is very interesting and deserves to be published with these revisions:

1) Since the activity of SIRT1 in tables 1, 2, and 3 is always "not applicable" except for S-carvone, this information could be put in the caption.

2) It would also be helpful to have in the tables the doses used in the model that worked to raise SIRT1 expression.

3) In Figure 1, among the beneficial effects mediated by the increase of sirt1, the reproductive system should be mentioned.

Author Response

We appreciate the opportunity to revise the manuscript and thank the reviewer for the helpful suggestions.

1) Since the activity of SIRT1 in tables 1, 2, and 3 is always "not applicable" except for S-carvone, this information could be put in the caption.

R: We removed the corresponding column from tables 2 and 3 and changed the caption to indicate that each table presents monoterpenes that increase SIRT1 expression. We couldn’t find a suitable expression to indicate that only the effect on SIRT1 expression, not activity, was evaluated for most monoterpenes, except S-(+)-carvone. Therefore, we didn’t make any changes to table 1, but would appreciate any suggestion from the reviewer.

2) It would also be helpful to have in the tables the doses used in the model that worked to raise SIRT1 expression.

R: The range of doses/concentrations of each compound tested was included in each table.

3) In Figure 1, among the beneficial effects mediated by the increase of sirt1, the reproductive system should be mentioned.

R: Effects of SIRT1 on the male and female fertility were introduced in fig. 1 and the graphical abstract

Reviewer 2 Report

Overall assessment:

The presented manuscript titled “Monoterpenes as Sirtuin-1 activators: therapeutic potential in aging and related diseases” presents the role of non-polyphenolic  monoterpenoids  as inducers of SIRT1 expression and activity in regards to their potential  in the treatment of age-related diseases.

The highest value of the study is that the data gathered in the manuscript documenting the role of natural, non-polyphenolic compounds that could become the main active ingredients in medicinal products acting as SIRT modulators. As authors underlie the readers can find there broadly discussed controversies and hurdles that still persists about SIRT1 and its potential to slowdown  aging and age-related diseases. The advantage of the study is detailed description of monoterpenes and  iridoids mechanism of action documented by well selected literature. The authors proved that taking into consideration the beneficial effects of those substances in animal studies they could be potentially used as novel pharmacological strategies in the treatment of subclinical inflammation being the precursor of metabolic diseases as well as utilizing their mechanism of avoiding the accumulation of non-metabolized deposits in neurogenerative diseases in humans, after further studies.

Finally the paper is extremely worth publishing in Biomolecules

Author Response

We thank the reviewer for the positive comments.

Reviewer 3 Report

This review is interesting and well-written.

I have only one concern (which is important according to my opinion):

While the authors described evidence supporting the potential of non-polyphenolic compounds (and their relationship with sirtuin 1) as pharmaceuticals for therapeutic application in age-related diseases, they 

did not discuss the potential of these compounds to lead to cellular damage.

In particular, it should be discussed also that both polyphenols and other compounds (terpenoids and others), as well as other activators of SIRT1 (e.g. exercise and caloric restriction), that can enhance SIRT1 levels, can also cause damage (probably in a dose-dependent manner) because they alter the oxidative and inflammatory homeostasis (via SIRT1 pathway and other mechanisms). Moreover, the consequences of altering the redox state in the body and the expression and activity of a stress sensor as SIRT1 should be also discussed, There are helpful articles in this field (e.g. Front Pharmacol. 2016 Feb 12;7:24. doi: 10.3389/fphar.2016.00024.; J Nutr Health Aging. 2019;23(3):246-250. doi: 10.1007/s12603-018-1149-7.; Nutrients. 2022 May 17;14(10):2092. doi: 10.3390/nu14102092.Br J Dermatol. 2021 Jun;184(6):1132-1142. doi: 10.1111/bjd.19666).

This issue is insufficiently addressed with few words in the discussion section!

Author Response

We appreciate the opportunity to revise the manuscript and thank the reviewer for the pertinent and constructive comments.

We agree with the reviewer’s view that some counterarguments were lacking in our manuscript. Thus, we included a discussion about hormetic effects of mild oxidative stress on SIRT1 and their potential disruption by monoterpenes. We also added a paragraph highlighting non-coding RNAs as modulators of SIRT1 expression and their potential as targets for monoterpenes. Finally, we discuss potential implications and risks of increasing SIRT1 expression and activity and their relevance for the potential therapeutic use of monoterpenes.